# First Report on Genome Analysis and Pathogenicity of *Vibrio tubiashii* FP17 from Farmed Ivory Shell (*Babylonia areolata*)

**Chen Dai [1,2], Xiaoxin Li [1], Dapeng Luo [1,3], Qingming Liu [1], Yun Sun [4], Zhigang Tu [1,2,3,*] and Minghui Shen [1,2,3,*]**

1   Hainan Provincial Key Laboratory of Tropical Maricultural Technologies, Hainan Academy of Ocean and Fisheries Sciences, Haikou 571126, China
2   Key Laboratory of Utilization and Conservation for Tropical Marine Bioresources of Ministry of Education, Hainan Tropical Ocean University, Sanya 572022, China
3   Hainan Key Laboratory for Conservation and Utilization of Tropical Marine Fishery Resources, Yazhou Bay Innovation Institute, Hainan Tropical Ocean University, Sanya 572022, China
4   State Key Laboratory of Marine Resource Utilization in South China Sea, Hainan University, Haikou 570228, China
*   Correspondence: tuzg@hnhky.cn (Z.T.); smh112266@aliyun.com (M.S.)

**Abstract:** Ivory shell (*Babylonia areolata*) is a commercially important aquaculture species mainly found on the southeast coast of China. However, it has been greatly affected by vibriosis in recent years. In this study, FP17 (a potential pathogen) was isolated from a dying ivory shell with "acute death syndrome" and confirmed as a pathogen via infectious experiment. Furthermore, phylogenetic analysis based on the average nucleotide identity (ANI) sequencing of the 16S rRNA gene and housekeeping genes (*ftsz*, *gapA*, *gyrB*, *mreB*, *pyrH*, *rpoA*, and *topA*) indicated that FP17 was identical to *Vibrio tubiashii*. Transmission electron microscopy showed that FP17 is curved and has a short rod shape, with a single flagellum. Besides, the calculated $LD_{50}$ after the intramuscular injection of FP17 was $2.11 \times 10^6$ CFU/g at 14 d. The genome of the FP17 strain consists of two chromosomes and one plasmid with 5,261,336 bp and 45.08% GC content, including 4824 open reading frames (ORFs) and 150 non-coding RNAs (ncRNA). Genome mining revealed that 120 candidate gene clusters, including vibrioferrin and flagellum-related proteins, are responsible for virulence. Comparative genomic analysis showed that vibrioferrin genes, such as *pvs* and type VI secretion system protein genes (*vas*), are specific in *V. tubiashii* FP17 but not in the ATCC19109 strain. Furthermore, 92 antimicrobial resistance (AMR) genes, such as *tufA*, *tet(35)*, *crp*, etc., were mapped within the genome as the potential candidate for virulence, consistent with antibiotic susceptibility assay. This is the first study to describe the complete genome sequence of *V. tubiashii* infecting ivory shell. The genetic characteristics, virulence factors, and antimicrobial resistance of the *V. tubiashii* strain FP17 were also explored.

**Keywords:** *Babylonia areolata*; *Vibrio tubiashii*; whole genome sequencing; pathogenicity; virulence factors

## 1. Introduction

Ivory shell, *Babylonia areolata* (Mollusca, Gastropoda, Buccinidae, *Babylonia*), belongs to carnivorous gastropods and is mainly found in the sandy bottom of the tropical and subtropical sea areas of the Indo-Pacific Ocean (depth 5–20 m) [1]. *B. areolata* is the most promising economically cultivated shellfish in the southeastern coastal provinces (Hainan, Guangdong, and Fujian) of China. However, the recent outbreak of acute death diseases has significantly affected the *B. areolata* industry, leading to a loss of at least 40% of annual output [2].

Acute death syndrome has a high progression rate and high mortality. Besides, ivory shells with this disease cannot eat and drill into the sand, leading to death within days. Nevertheless, only a few studies have assessed the disease and indicated that *Vibrio tubiashii* is the pathogen of *B. areolata* disease, and not *V. harveyi* [2,3].



*V. tubiashii* is a Gram-negative bacterium, causing bacillary necrosis. Besides, *V. tubiashii* is associated with high larval and juvenile bivalve mollusk mortality in hatcheries [4]. *V. tubiashii* has been isolated as a pathogen from multiple species, including *Crassostrea gigas* [5–7], *Mercenaria mercenaria* [4], *Hippocampus erectus* [8], *Haliotis tuberculata* [5], *Pinna nobilis* [9], *Oncorhynchus mykiss* [10], and *Crustose coralline* algae [11]. However, a few misidentifications occurred before the first report of the complete genome sequence [12], including strains RE22, RE98, and ATCC19105 [13–16]. A draft genome sequence of ATCC19106 has been isolated from oyster spat from *Crassostrea virginiaca*, and it consists of two chromosomes with 5,353,266 bp (4864 protein-coding and 86 RNA genes). The multi-antibiotic resistance protein MarC is associated with kanamycin resistance [17]. The complete genome sequence of ATCC19109 has been isolated from the juvenile hard clam, *M. mercenaria*, and it contains two chromosomes (3,294,490 and 1,766,582 bp), two megaplasmids (251,408 and 122,808 bp), and two plasmids (57,076 and 47,973 bp) [12,18]. Therefore, additional evolutionary comparison is needed between the strains crossing infected species. Although *V. tubiashii* was discovered 50 years ago, its pathogenesis requires further analysis. Besides, virulence factors, antimicrobial resistance (AMR) genes, and other genomic analyses should be further studied. However, studies have shown that hemolysin and protease are the main virulence factors of the disease [5–7].

In this study, a novel strain of *V. tubiashii* (FP17) was isolated from moribund *B. areolata* farmed at Hainan during the last two years. The FP17 strain was confirmed to be the etiological agent of the acute death syndrome of *B. areolata*. The complete genome sequence of *V. tubiashii* was then characterized and analyzed, and the potential virulence genes and AMR sequences were predicted using bioinformatic approaches. Furthermore, antibiotic susceptibility was assessed. Drug-paper diffusion assay showed that FP17 was resistant to tetracyclines and sensitive to quinolones. Therefore, this study provides insights into the pathogenesis, prevention, and treatment of *V. tubiashii*-associated diseases.

## 2. Materials and Methods

### 2.1. Ethics Statement

All animal experiments were approved by the Ethics Committee of the Hainan Academy of Ocean and Fisheries Sciences (protocol code HAOFS-2021-0012 and approved on 9 August 2021).

### 2.2. Disease Outbreak Description and Bacterial Isolation

Epidemic disease was detected in 90–100-day-old *B. areolata* post-hatching in an ivory shell industrial unit, which infected about 80% of *B. areolata* in the farming pond. The average water temperature and salinity were 29 °C and 30 ppt, respectively. In this study, 20 typically infected and dying *B. areolata* were randomly selected from the affected ivory shell population for bacteriological examination.

The bacterium was first obtained from the visceral masses of ivory shells (*B. areolata*). Briefly, wet visceral mass (0.5 g) was homogenized with 1 ml of sterilized saline water. The homogenate (50 µL) was then plated on plates with 2216E agar (Hopebio, Qingdao Hope Bio-Technology Co., Ltd., Beijing, China) and Thiosulfate Citrate Bile Sucrose agar (TCBS) (Beijing Land Bridge Technology Co., Ltd., Beijing, China) and incubated at 30 °C for 24 h. The single dominant colony was re-streaked onto Tryptic Soy Agar (TSA) (BD Difco, Sparks, MD, USA) and Luria–Bertani (LB) Agar plates with 1.5% NaCl, then cultured. A single colony was then cultured in LB medium containing 1.5% NaCl for 16–18 h. The bacterial morphology was determined using transmission electron microscopy (TEM) (HT7700, Hitachi, Japan).

### 2.3. Infection Experiment

The isolated FP17 strain was cultured in 2% NaCl LB at 30 °C and 180 rpm overnight. The bacterial pellet was centrifuged, washed using $1\times$ phosphate buffered saline (PBS), and suspended in PBS. The healthy three-month-old snails were obtained from Wenchang

with a shell length and weight of $3.62 \pm 0.22$ g and $2.64 \pm 0.27$ cm, respectively. Each group had 10–20 snails. The snails were kept in water containing dissolved oxygen (DO) ($7.5 \pm 0.4$ mg/L) at a temperature, pH, and salinity of $28.8 \pm 1.6$ °C, $8.1 \pm 0.1$, $30.5 \pm 0.5$ ppt, respectively. Bacterial suspension (100 μL) with $1.8 \times 10^8$, $1.8 \times 10^7$, $1.8 \times 10^6$, and $1.8 \times 10^5$ CFU/mL, was intramuscularly injected, respectively, to each group. Phosphate buffered saline was used as a negative control. The snails were fasted after infection. The number of deaths of *B. areolata* was recorded every 2 h (the first day) post-infection for 14 d. Bacteriological analysis of dead snails was also conducted in all cases. Death was only considered to be caused by the inoculated bacterium when the original strain used for inoculation was re-isolated from the visceral masses of the inoculated snail in pure culture. The $LD_{50}$ value was calculated as previously described [19].

### 2.4. Biochemical and Physiological Characterization

The growth condition with 1%, 3%, 6%, 8%, and 10% NaCl was assayed using peptone broth (Guangdong Huankai Microbial Sci. & Tech. Co., Ltd., Guangzhou, China). Additional phenotypic characteristics were determined using API 20E (bioMérieux, Craponne, France) and Vibrioceae Bacterial Biochemical Identification Kit (Guangdong Huankai Microbial Sci. & Tech. Co., Ltd., China) following the manufacturer's instructions. Hemolysis of red cells was determined using Columbia blood agar (Guangdong Huankai Microbial Sci. & Tech. Co., Ltd., China).

### 2.5. DNA Extraction for the Whole Genome Sequencing

Genomic DNA (gDNA) of FP17 was extracted from the single culture suspension with an E.Z.N.A.® Bacteria DNA Kit (Omega Bio-Tek Inc., Norcross, GA, USA) following the manufacturer's instructions. Quality control of the purified DNA sample was also performed. The gDNA was quantified using TBS-380 fluorometer (Turner BioSystems Inc., Sunnyvale, CA, USA). High quality DNA sample (>6 μg) was used to construct fragment library. The genome was sequenced by a commercial service (Shanghai BIOZERON Co., Ltd., Shanghai, China) using a combination of PacBio RS II platform (Pacific Biosciences, Menlo Park, CA, USA) and Illumina novaseq6000 platforms (Illumina, San Diego, CA, USA) on single molecule real-time (SMRT) cells of PacBio system. The Illumina data were used to evaluate the complexity of the genome and correct the PacBio long reads.

### 2.6. Genome Assembly, Annotation and Analysis

The raw paired-end reads were trimmed and quality was controlled based on trimmomatic parameters. Clean reads were used for further analysis. The genome was optimized and assembled using ABySS based on multiple-Kmer parameters. Then, Canu was used to assemble the PacBio-corrected long reads. Finally, GapCloser software was used to fill the remaining local inner gaps and correct the single-base polymorphism for the final assembly results. The Ab initio prediction method was used to get gene models for FP17. Gene models were identified using GeneMark [20]. All gene models were then blasted against the non-redundant (NR in National Center for Biotechnology Information, NCBI) database, SwissProt (http://uniprot.org, accessed on 10 Februery 2022), Kyoto Encyclopedia of Genes and Genomes (KEGG) (http://www.genome.jp/kegg/, accessed on 6 March 2022), and Clusters of Orthologous Genes (COG) (http://www.ncbi.nlm.nih.gov/COG, accessed on 10 March 2022) for functional annotation via blastp module. In addition, tRNAs were identified using the tRNAscan-SE (v1.23, http://lowelab.ucsc.edu/tRNAscan-SE, accessed on 5 April 2022), while rRNAs were determined using the RNAmmer (v1.2, http://www.cbs.dtu.dk/services/RNAmmer/, accessed on 10 April 2022). The putative virulence factor of the FP17 strain was predicted in the Virulence Factor Database (http://www.mgc.ac.cn/VFs/main.htm, accessed on 7 May 2022) via diamond blastp software (the parameter is evaluated 0.00001 –id 70 –subject-cover 90). Antimicrobial resistance (AMR) genes were identified through BLAST search of the Comprehensive Antibiotic Resistance Database (https://card.mcmaster.ca/, accessed on 15 May 2022). Circular genome

maps were generated using the CGView Server based on the information generated by the genome annotation [21].

### 2.7. Phylogenetic Analyses

Nine phylogenetic tree analyses were performed based on average nucleotide identity (ANI), 16S rRNA gene, and housekeeping genes (*ftsz*, *gapA*, *gyrB*, *mreB*, *pyrH*, *rpoA*, and *topA*), as previously described [22,23].

### 2.8. Comparative Genome Analysis

The software cd-hit (version: 4.6.1, http://cd-hit.org, accessed on 19 May 2022) was used to cluster the protein sequences of ATCC19109 and FP17 for core and specific genes. The parameters were set as identity ($\geq$0.4) and alignment length ($\geq$longer sequence $\times$ 0.4). The orthologous genes of the genomes were generated using PhyML by comparing them to several other closely related species maintaining high housekeeping gene similarities. The variation of functional genes between *V. tubiashii* ATCC19109 and FP17 was also compared. MUMmer was used (version: 3.23, http://mummer.sourceforge.net/, accessed on 21 May 2022) to compare and determine the large-scale collinearity between the genomes. LASTZ was used (version: 1.03.54, http://www.bx.psu.edu/miller_lab/dist/README. lastz-1.02.00/, accessed on 23 May 2022) to compare the regions and confirm the partial position arrangement relationship, and find the areas of translocation (Translocation/Trans), inversion (Inversion/Inv), and translocation + inversion (Trans + Inv).

### 2.9. Antibiotic Resistance Profiling

Antibiotic susceptibility was tested using the Kirby–Bauer disk diffusion method [24]. Single colony of FP17 was cultured in 2% NaCl LB medium at 180 rpm until $OD_{600} \approx 0.8$. Bacterial suspension (200 μL) was spread uniformly on 2% NaCl Mueller Hinton agar plates (Hangzhou microbial reagent Co., Ltd., Hangzhou, China). The plates supplemented with antibiotic disks were incubated at 30 °C overnight. An inhibition zone diameter was measured as an indicator of sensitivity following the manufacturer's criteria.

### 2.10. Statistical Analysis

All statistical analyses were performed using GraphPad PRISM 9.0 software (San Diego, CA, USA). Results are expressed as the mean $\pm$ standard error of the mean (SEM).

## 3. Results

### 3.1. Phenotypic Characteristics

The FP17 strain was isolated from moribund *B. areolata* with acute death syndrome, exhibiting clinical signs of shell turnover and fast-developing death. The purified colony was Gram-negative, and was grown on TCBS, TSA, and LB plates at 18~37 °C. The colonies on TSA or LB plate showed a greyish-white, opaque, and smooth-edged mucoid shape. The TEM showed a curved, rod-shaped capsuled bacterium of about 1.6–1.9 μm in length, 0.8–1 μm in width, and a 3–3.4 μm single flagellum. The FP17 strain was able to produce oxidase and unable to grow in the absence of NaCl (Table 1). The FP17 strain was positive for citrate utilization, nitrate reduction, and sucrose, while it was negative for arginine dihydrolase, lysine decarboxylase, ornithine decarboxylase, $H_2S$ production, urease, gelatinase production, tryptophan deaminase, indole production, and Voges–Proskauer, as well as glucose, mannose, inositol, sorbitol, rhamnose, melibiose, and arabinose. Additionally, yellow colonies and the beta-haemolysis of red cells were observed on TCBS plate and Columbia blood agar, respectively. The phenotypic characteristics of the reference *V. tubiashii* and ATCC19109 strains are shown in Table 1 [25,26].

**Table 1.** Phenotypic characteristics of the *Vibro tubiashii* FP17.

| Characteristic | FP17 | ATCC19109 [25] | *V. tubiashii* [26] |
|---|---|---|---|
| Shape | Curved rods | Curved rods | Curved rods |
| Motility | + | + | + |
| Polar flagella | + | + | + |
| Gram staining | − | − | − |
| Grow at 1%, NaCl | + | + | + |
| 3%, | + | + | + |
| 6%, | + | + | + |
| 8% | − | −/+ | −/+ |
| 10% | − | − | − |
| β-galactosidase | − | − | −/+ |
| Arginnine dihydrolase | − | + | −/+ |
| Lysine decarboxylase | − | − | − |
| Ornithine decarboxylase | − | − | − |
| Citrate utilization | + | + | + |
| $H_2S$ production | − | − | − |
| Urease | − | − | − |
| Tryptophan deaminase | − | + | N |
| Indole production | − | + | + |
| Voges–Proskauer | − | − | − |
| Gelatinase production | − | + | + |
| Glucose | − | − | + |
| Mannose | − | − | + |
| Inositol | − | − | − |
| Sorbitol | − | − | − |
| Rhamnose | − | − | − |
| Sucrose | + | + | + |
| Melibiose | − | − | − |
| Amygdalin | −/+ | N | N |
| Arabinose | − | − | − |
| Oxidase | + | + | + |
| Nitrate reduction | + | + | + |
| Growth on TCBS | Y | Y | Y |

+: positive; −: negative; −/+: variable; Y: colonies grow yellow on TCBS; N: no data.

### 3.2. Pathogenic Activity

Injections of the bacterial strain FP17 were lethal. At 2 h post-challenge, the ivory shell injected with FP17 ($10^8$ CFU/mL) died. FP17, as the dominant strain, could be re-isolated from the visceral masses of the moribund or dead ivory shell. The survival rate of the ivory shell infected with FP17 is shown in Figure 1. However, no mortality was reported in the control group. The calculated $LD_{50}$ at 14 d of intramuscular injection of FP17 was $2.11 \times 10^6$ CFU/g.

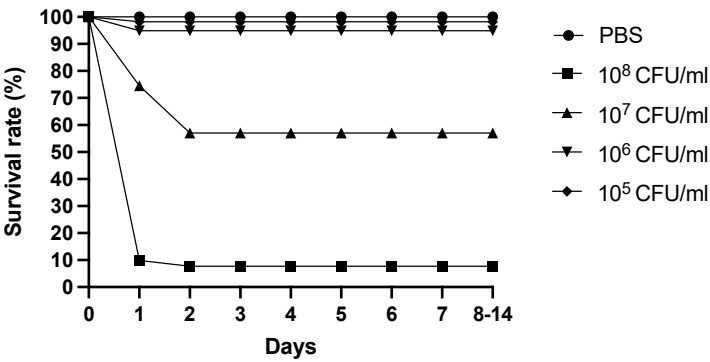

**Figure 1.** Survival of *B. areolata* challenged with different concentrations of FP17. Phosphate buffered saline was used as negative control.

### 3.3. Genome Characteristics

The fully assembled genome of FP17 contains two closed chromosomes and one closed plasmid (NCBI accession no.: PRJNA792890) with 5,261,336 bp and 45.08% GC content, consisting of chromosome I (N50 length = 3,300,914 bp with 45.09% GC), chromosome II (1,649,311 bp with 44.67% GC), and one plasmid (pFP31) (311,111 bp with 47.27% GC). Chromosome I contains 2984 predicted genes, 111 tRNAs, and 31 rRNAs (10 23S rRNA, 10 16S rRNA, and 11 5S rRNA). Chromosome II contains 1569 predicted genes and 8 tRNAs, while pFP31 only contains 271 predicted genes. Circular genomes were individually mapped, as shown in Figure 2. The predicted genes were further grouped into COG functional categorization (4824 OFRs), including 706 functionally unknown genes (Figure 3A). The dominant groups included 412 genes in class K (transcription), 327 genes in class E (amino acid transport and metabolism), 326 genes in class T (signal transduction mechanisms), 298 genes in class P (inorganic ion transport and metabolism), and 277 genes in class M (cell wall/membrane/envelope biogenesis).

Most genes that are essential for bacterial growth and mobility, such as the genes related to flagellum (*flgA/B/C/D/E/F/G/H/I/K/L/O/P/T*), are located in chromosome I. The co-localization of the genes of *dnaA*, *dnaN*, *recF*, and *gyrB* was identified as the replicative origin in chromosome I, similar to *V. chloerae* [27]. The *dsdA*, *thrS*, and genes encoding ribosomal proteins (*L20* and *L35*) are located on chromosome II, while the corresponding genes are found on chromosome I in *V. parahaemolyticus* [28]. Quorum-sensing genes exist on both the chromosomes of FP17. For example, *luxO/P/Q/S/U*, belonging to the autoinducer-2 (AI-2) quorum-sensing mechanism, is split into *luxO/S/U* on chromosome I and *luxP/Q* on chromosome II. Also, the *ada-alkA*, *sbcC/D*, and *dcm* genes are found on chromosome II.

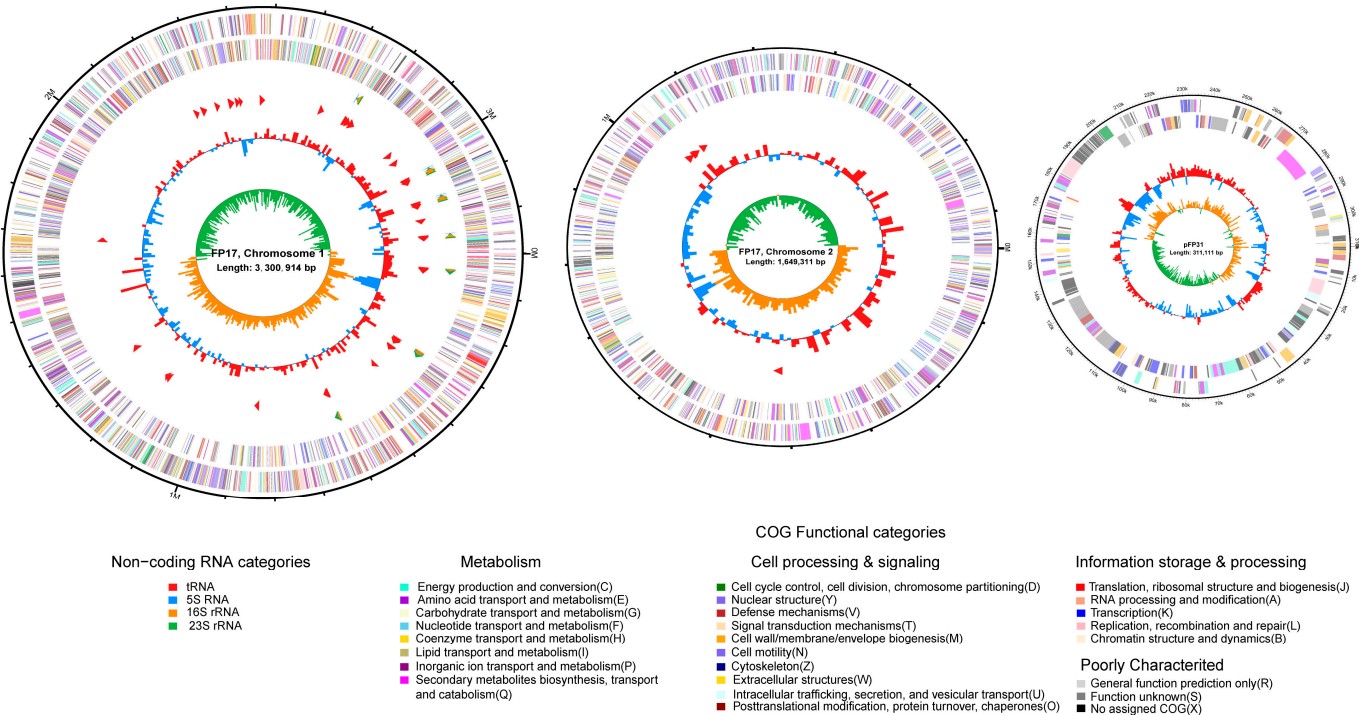

**Figure 2.** Genome map of *V. tubiashii* FP17. Chromosomes I, II, and plasmid pFP31 were mapped based on their genomic size. The outermost circle is the identification of the genome size; the second circle and the third circle are the CDS on the positive and negative strands, and the different colors indicate the functional classification of the COG of the CDS; the fourth circle is rRNA and tRNA; the fifth circle is the GC content; the innermost circle is the GC skew value.

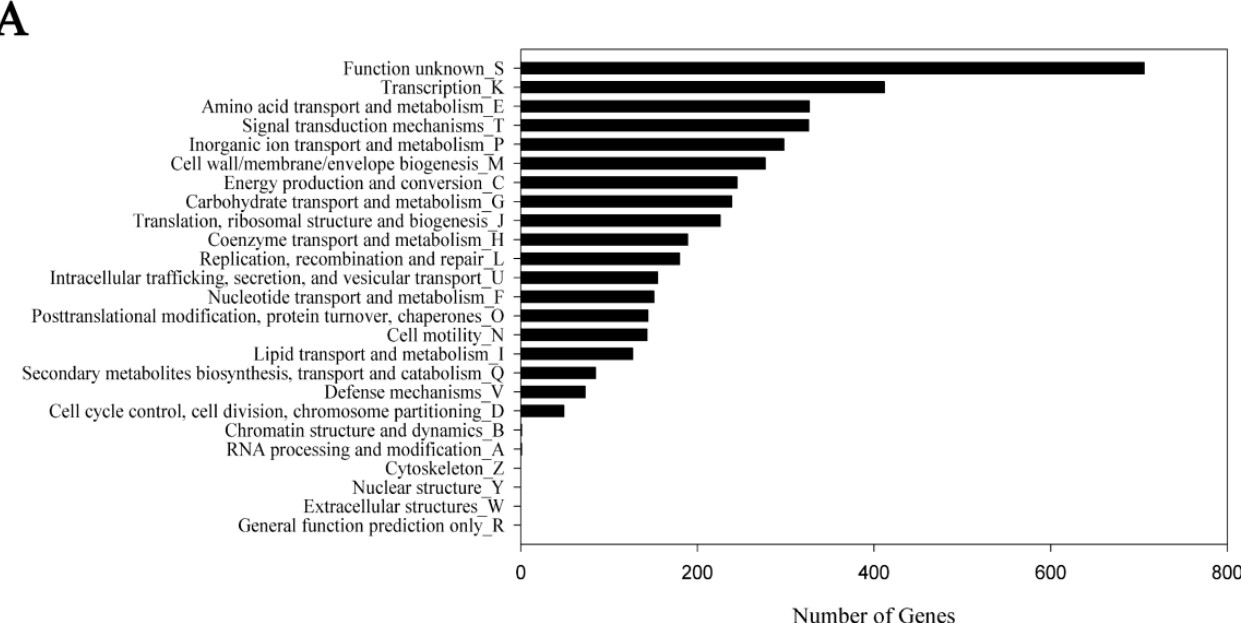

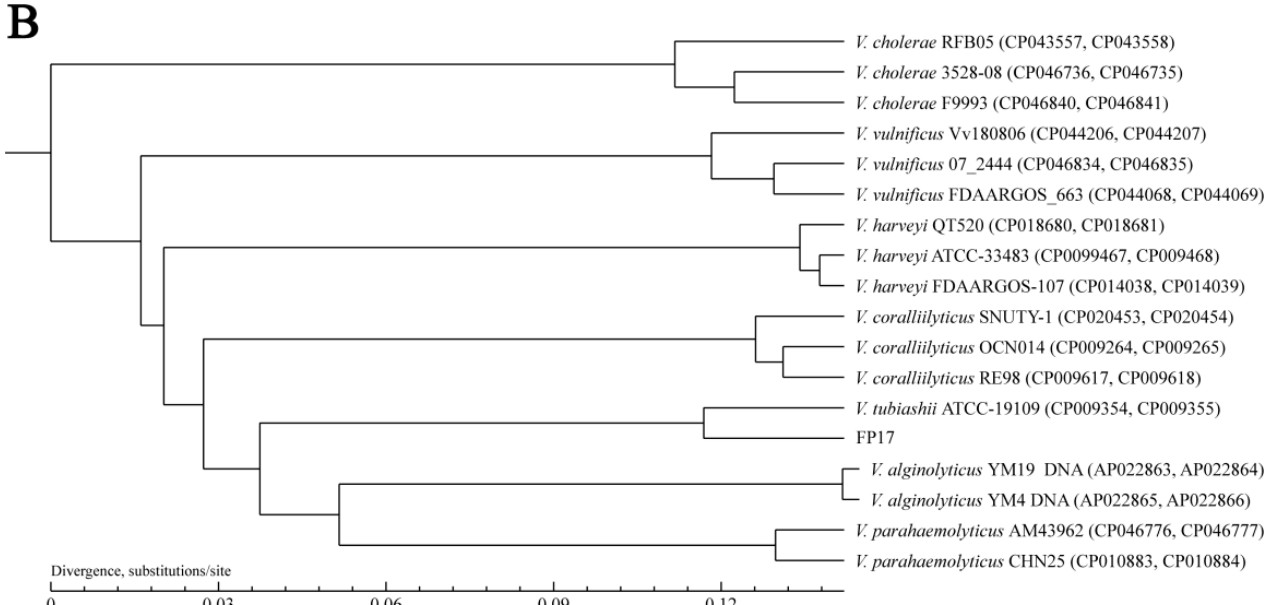

**Figure 3.** (**A**) Functional categorization of *V. tubiashii* FP17 based on the COG database. (**B**) The phylogenetic tree of *V. tubiashii* FP17 with closely related vibrio species according to the ANI value.

### 3.4. Phylogenetic Tree Analysis

The phylogenetic tree analysis was conducted based on ANI, the sequencing of the 16S rRNA gene, and the housekeeping genes (*ftsz*, *gapA*, *gyrB*, *mreB*, *pyrH*, *rpoA*, and *topA*) among 17 *Vibrio* strains (Figures S1 and S2). Phylogenetic analysis showed that FP17 is the most closely related to *V. tubiashii* ATCC19109 (Figure 3B).

### 3.5. Comparative Genome Analysis

Unlike *V. tubiashii* ATCC19109, which contains four plasmids, FP17 only has a unique plasmid (Figure 2). Furthermore, the in-depth comparison of the FP17 genome with the ATCC19109 strain showed several rearrangements (Figure 4 and Table S1). A total of 3915 core genes are shared by FP17 and ATCC19109. Each of them has 748 and 897 specific genes, respectively (data not shown). Twenty virulence factor genes and six AMR genes, including vibrioferrin-related genes (*pvsA/B/C/D/E*, *pvuA/C/D/E*, and *psuA*) and type VI

secretion system genes (*vasA/B/C/D/E/F/G/K*), were specifically found in FP17 but not in ATCC19109 [12].

**Figure 4.** Collinear pairwise comparison of the *V. tubiashii* FP17 and ATCC19109 strain.

*3.6. Virulence Factors*

The VFD sequence blasting was further performed for virulence factor genes. The 120 virulence factors of *V. tubiashii* FP17 are summarized in Table 2. In addition, general functional genes, such as flagellin and flagellar motor protein, which are important for bacterial colonization, vibrioferrin, the secretion system, and chemotaxis protein were found in chromosome I.

The type II secretion system (T2SS), type IV secretion system (T4SS), and type VI secretion system (T6SS) were also detected in the FP17 genome. T6SS was split into two chromosomes (T6SS1 and T6SS2 located on chromosomes I and II, respectively), while the other secretion system genes were only found in chromosome I. Hemolysin co-regulatory protein (Hcp), is a T6SS translocation protein with a six-membered ring protein that can be stacked in the periplasm to form a long tube [29,30]. In this study, T3SS was not detected in the whole genome of FP17.

*pvsA/B/C/D/E* and *psuA-pvuA/C/D/E* are operons crucial for the biosynthesis and transport of the siderophore vibrioferrin, and were found in the FP17 genome [2]. Gene *psuA* and *pvuA* are iron membrane-bound receptors that can bind the ABC transport system and transport the siderophore complex. The *PvuBCDE* system is located in the inner membrane, which transports iron-vibrioferrin to the inner membrane [31]. The ferric uptake regulator, *fur*, the repressor of iron uptake systems under the iron sufficient condition, was also identified in the FP17 genome [31,32]. *hap/vvp*, as hemagglutinin protease, was also found in the FP17 genome. VVP was originally found in *V. vulnificus*, and has a vascular permeability-enhancing function. *hap* is the *V. cholera* hemagglutinin protease [33]. In this study, VVP and *hap* did not appear in database blasting, but *tlh* and *tdh*, the common haemolysin toxins, were found in the FP17 genome.

**Table 2.** Virulence factors of *V. tubiashii* FP17.

| VF Groups | VF Genes | Annotation | Chromosome | Location |
|---|---|---|---|---|
| Adherence | *mshA, mshE, mshG, mshL, pilD, pilT, pilU* | Fimbrial adhesin | Chromosome I | 353,050–445,936, 2,854,510–2,862,684 |
| | *rcpA, tadA* | Fimbrial adhesin | Chromosome II | 724,950–726,275, 727,929–729,194 |
| | *Fphi_1039, htpB, IlpA, VP1611* | Non-fimbrial adhesin | Chromosome I | 2,916,318–2,917,502, 173,880–175,526, 2,472,897–2,473,706, 1,535,525–1,538,158 |
| | *gbpA* | Non-fimbrial adhesin | Chromosome II | 1,158,586–1,160,046 |
| Metabolic factor | *psuA, pvsA, pvsB, pvsC, pvsD, pvsE, pvuA, pvuC, pvuD, pvuE* | Iron uptake (Siderophore uptake system) | Chromosome I | 384,230–399,756 |
| | *hemB, hemE, hemL* | Iron uptake (Heme uptake system) | Chromosome I | 3,207,563:3,208,606, 2,950,897:2,951,964, 513,692–514,984 |
| | *vctA, vctC, vctD, vctG* | Iron uptake (Ton system/ABC transporter system) | Chromosome II | 987,568–994,305 |
| Secretion | *epsE, epsF, epsG,gspD* | Type II secretion system | Chromosome I | 73,629–78,853 |
| | *clpB/vasG, clpV, evpJ, icmF/vasK, vasA, vasB, vasD, vasE, vasF, VCA0109, vipA/mglA, vipB/mglB,* | Type IV secretion system | Chromosome I | 600,132–602,705, 1,023,378–1,042,532 |
| | *hcp-2* | Type VI secretion system | Chromosome II | 417,585:418,103 |
| Motility | *cheA, cheB, cheR, cheV, cheW, cheY, cheZ, flaA, flaB, flaC, flaD, flaE, fleN, flgB, flgC, flgD, flgE, flgF, flgG, flgH, flgI, flgK, flgL, flgO, flgP, flgT, flhA, flhB, flhF, fliA, fliE, fliF, fliG, fliI, fliJ, fliM, fliN, fliP, fliQ, fliR, flmH, flrA, flrB, flrC, motA, motB, motX, motY* | Flagella-mediated motility | Chromosome I | 245,246–245,995, 881,531–995,408, 23,33,278–2,385,045 |
| Immune modulation | *rfaD, rmlA, rmlB, rmlC, wbfY, wbfU* | Antiphagocytosis | Chromosome I | 132,236–162,322, |
| | *ABBFA_003457* | Antiphagocytosis | Chromosome II | 695,532–696,542 |
| | *mrsA/glmM, pgi* | Exopolysaccharide | Chromosome I | 531,610–532,950, 2,900,478–2,902,130 |
| | *gmhA/lpcA, kdsA, lpxC* | Inflammatory signaling pathway | Chromosome I | 2,352,195–2,597,372 |
| | *PLES_19091* | Inflammatory signaling pathway | Chromosome II | 680,579–681,859 |
| Exoenzyme | *hap/vvp,* | Hemagglutinin protease | Chromosome II | 225,080–226,903 |
| Biofilm | *luxS* | Quorum sensing | Chromosome I | 2,759,812–2,760,330 |
| Stress/survival | *katB* | Catalase | Chromosome II | 594,195–596,372 |
| Regulation | *fur* | Ferric uptake regulator | Chromosome I | 854,626–855,075 |
| Other | *huvB, huvC, huvX, huvZ* | | Chromosome II | 471,081–476,422 |

### 3.7. Genomic Islands (GIs) and Prophage

The genomic islands acquired from the horizontal transmission were also predicted in FP17 according to the size, the frequency of interaction with genes encoded by tRNA, and the different GC content compared with other regions of the genome [34]. Many sub-categories were identified based on the multiple functions, such as the pathogenicity island (PAI) (containing virulence genes) and antibiotic resistance gene islands (containing many AMR genes). A total of 23 PAIs were predicted in the genome of the FP17 strain, of which 16 PAIs were located in chromosome I, 5 in chromosome II and 2 in plasmid1 (Figure 5). A total of 259 and 77 genes were mapped in PAIs of chromosomes I and II, respectively. The gene profile of the PAIs was different between the two chromosomes, except for *zot* (zonular occludens toxin), which was equally presented in three PAIs of both chromosomes. Chromosome I contained several enzymes, such as *waaA/F/L* (the function associated with LPS and the biosynthesis of flagellum), *paaK* (phenylacetic acid ligase), and *rfbA/B/C/D* (involved in rhamnose synthesis and LPS embedding). The tol-pal system genes and with *ybgC*, necessary for the morphology and virulence of Gram-negative bacteria, were found on chromosome I. Additionally, the adhesin gene (*alpA*) and *Neisseria meningitides* adhesin A (*nadA*) were mapped for the bacterial growth and immune response in chromosome I.

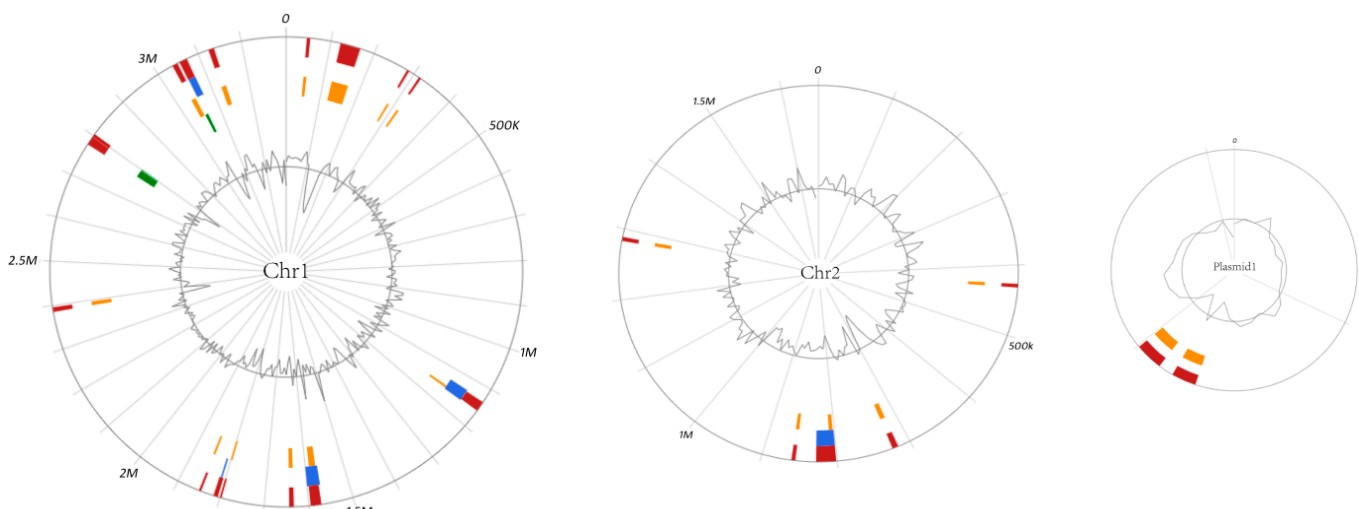

**Figure 5.** Genomic islands prediction in *V. tubiashii* FP17 by Islandviewer. The inner circle curve shows the distribution of GC content. Red indicates the integration interval of different gi prediction methods; Green means gi area predicted by islandPath-DIMOB method; Orange means gi area predicted by SIGI-HMM method; Green means gi area predicted by islandPick method.

Temperate bacteriophages that integrate into the host genome are known as prophages. Prophage sequences allow some bacteria to acquire antibiotic resistance, enhance adaptation to the environment, and improve adhesion or make bacteria pathogenic. The prophage sequences of FP17 were analyzed using PHAST (Table S2). Chromosome I contained one intact and one incomplete prophage sequence. Chromosome II and pFP31 had one intact prophage sequence each. The schematic diagram of the prophage is shown in Figure 6.

### 3.8. Antibiotic Susceptibility

Ninety-two AMR genes, such as *tufA*, *tet(35)*, and *crp*, were identified. The genes are crucial for the resistance of elfamycin, tetracycline, and multiple drug resistances. These results are consistent with the antibiotic susceptibility testing data (Table S3). The *tet(35)* gene had an 88.1% identical rate and may be highly involved in tetracycline resistance.

A total of 34 antibiotics (for humans or veterinary medicines) were tested in parallel to explore antibiotic sensitivity (Table S4). The FP17 was sensitive to 9 antibiotics, moderately sensitive to 7 antibiotics, and resistant to 18 antibiotics. Among those antibiotics, chloramphenicol, fluofenicol, norfloxacin, carbenicillin, and ciprofloxacin were the most sensitive ones. However, chloramphenicol and other antibiotics cannot be used in the aquaculture industry, and thus only florfenicol and ciprofloxacin should be used.

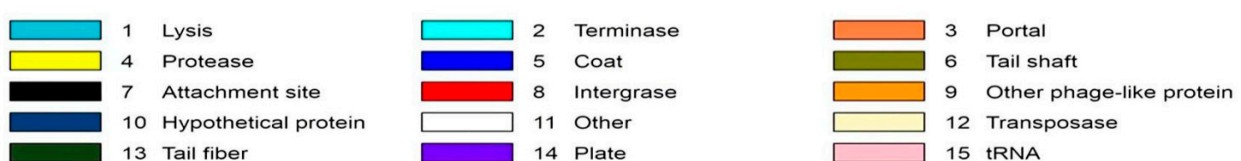

**Figure 6.** Schematic diagram of the FP17 prophage organization.

## 4. Discussion

In this study, the predominant bacterial strain FP17 was isolated from moribund *B. areolata* in an episode of mortality. Pathogenicity tests showed that FP17 had high virulence, and the symptoms of the diseased ivory shell were the same as naturally infected ivory shell. FP17 was quite similar to *V. tubiashii* based on Bergey's Manual of Determinative Bacteriology and ATCC19109, except for indole production, gelatinase production, and tryptophan deaminase [25,26]. However, it is not surprising that different biotypes or

serotype strains had different biochemical characteristics. For example, *V. tubiashii* HEL-5 was negative for gelatinase production, and tryptophan deaminase [8,26]. Molecular diagnostics are a precise and accurate method for bacterial identification at the species level [35]. In this study, phylogenetic tree analysis based on the sequences of the 16S rRNA gene and housekeeping genes (*ftsz*, *gapA*, *gyrB*, *mreB*, *pyrH*, *rpoA*, and *topA*) showed that FP17 had the highest homologies with *V. tubiashii* ATCC19109. Furthermore, FP17 could be assigned to the species *V. tubiashi* clustered with *V. tubiashii* ATCC19109 based on ANI. Therefore, the etiological agent of ivory shell mortality was *V. tubiashii*.

The acute death syndrome of ivory shells is associated with high mortality in *B. areolata* aquaculture. *B. areolata* with disease do not eat and cannot drill into the sand, and thus may die within days. The 14 d $LD_{50}$ of FP17 was $2.11 \times 10^6$ CFU/g body weight by intramuscular injection, indicating that it was moderately toxic to the ivory shell. In our study, a large number of deaths occurred between 2 and 24 h after challenge, and no deaths occurred after 48 h. *V. tubiashii* ATCC19106 and ATCC19109 were highly infectious to eastern oyster larvae, and may lead to death within 6 d ($LD_{50}$: $3.8 \times 10^3$ CFU/mL and $1.2 \times 10^4$ CFU/mL, respectively) [12]. *V. tubiashii* HEL-5 was pathogenic for seahorses (*Hippocampus erectus*) with $LD_{50}$ of $5.81 \times 10^5$ CFU/g fish weight. Furthermore, *V. tubiashii* HEL-5 caused deaths between 24 h and 72 h after treatment and no deaths after 144 h [8]. Therefore, although the pathogenic activity of *V. tubiashii* varied from isolated strains and infected hosts, the incubation period of FP17 in the ivory shell was shorter than that of the *V. tubiashii* strain in other host.

This is the first study to assess the complete genome of *V. tubiashii* isolated from moribund *B. areolata* with acute death syndrome. Unlike the reference *V. tubiashii* strain ATCC 19019, which contains two chromosomes and four plasmids, FP17 has two chromosomes and only one plasmid, indicating that FP17 may be an ancient strain based on bacterial evolution. The single plasmid of FP17 may have separated into two plasmids (p251 and p57) of ATCC 19109 during evolution. Herein, the possible virulence factors were predicted according to the molecular mechanisms of action of the assembled FP17 genome, mostly focusing on adherence, iron uptake, effective secretion system, motility, and metabolic factors. The results showed that flagellum-related genes were abundant in chromosome I of FP17.

Although iron is an essential element for the growth and reproduction of bacteria, most of the iron ions available in organisms exist as complexes, such as human transferrin, lactoferrin, and hemoglobin [36]. Bacteria can secrete low molecular weight siderophore chelating agents to chelate iron during infection invasion. The iron–siderophore complex binds to specific outer membrane receptors to complete iron transport across the membrane [37]. The TonB transduction system is required in Gram-negative bacteria [38]. Most *Vibrio* species contain two TonB systems (*TonB1-ExbB1-ExbD1* and *ExbB2-ExbD2-TonB2*) [39]. The transcription of the TonB systems is regulated by the iron concentration in the culture medium [40].

Genes *cyaB* (adenylate cyclase), *hlyB/D*, and *zot* were present in plasmid pFP31, indicating that these virulence factors were plasmid-associated. Interestingly, *hlyB/D* and *tolC*, composed of a specific membrane export complex, were found in the FP17 genome but not the secretion product hlyA (haemolysin) that depends on the complex. Several *lux* genes and *vas* genes may link FP17 to biofilm formation, the sensor quorum effector, and T6SS, which were verified in our further experiments (data not shown).

The genomic island is part of a genomic DNA region acquired by horizontal gene transfer, defined as the transfer of genetic material between bacterial cells without cell division [41]. Many type III and type IV secretion systems are located in genomic islands. The genome of FP17 contained PAI on both chromosomes (I and II), including 259 and 77 genes, respectively. Lipoprotein (lpp) was found in the PAI of chromosome II. Lipoprotein plays an envelope structural role and is important for bacterial pathogenesis [42].

FP17 also contained various AMR genes, including the *tet(35)* gene and the *bla2* gene, but not the *bla1* and *qnrA* genes, consistent with the antibiotic test. Genes *bla1* and *bla2*

encode functional beta-lactamases. *bla1* is a penicillinase, while *bla2* is a cephalosporinase gene. The resistant bacterial strains do not contain the genes responsible for the conferred resistance [43]. Although the antibiotic resistance analysis detected several AMR genes, only FP17 was highly sensitive to florfenicol and chloramphenicol, consistent with the results of another isolated *V. tubiashii* strain, HEL-5 [8]. Although FP17 was resistant to kanamycin, the genes of β-lactamases and MarC were not detected in FP17. Therefore, the resistance mechanisms of FP17 require further study.

## 5. Conclusions

A novel strain of *V. tubiashii*, FP17, was isolated from *B. areolata* in Hainan, China, after a large-scale acute death. Infectious experiment identified that FP17 was the pathogen causing the acute death syndrome in *B. areolata* with an $LD_{50}$ value of $2.11 \times 10^6$ CFU/g body weight.

The genome of the FP17 strain consisted of two chromosomes and one plasmid with 5,261,336 bp and 45.08% GC content, including 4824 open reading frames (ORFs) and 150 non-coding RNA (ncRNA). Genome mining revealed that 120 candidate gene clusters were responsible for the virulence factors, including vibrioferrin and flagellum-related proteins.

Comparative genomic analysis showed that *pvs* and *vas* were specific in *V. tubiashii* FP17 but not in ATCC19109.

Ninety-two AMR genes were identified and were related to resistance to elfamycin, tetracycline, and multiple drug resistances, consistent with our antibiotic susceptibility assay data.

**Supplementary Materials:** The following supporting information can be downloaded at: https://www.mdpi.com/article/10.3390/fishes7060396/s1, Figure S1: The neighbor-joining phylogenetic tree of *V. tubiashii* FP17 strain based on several housekeeping genes: 16S rRNA(A), *ftsz*(B), *gapA*(C), *gyrB*(D), *mreB*(E), *pyrH*(F), *rpoA*(G), and *topA*(H); Figure S2: Average nucleotide identity (ANI) value between 17 vibrio strains; Table S1: Comparison of the chromosomal properties of *V. tubiashii* FP17, ATCC 19019, and the ATCC19016 strain; Table S2: Information of prophage in FP17; Table S3: Summary part of the antimicrobial-resistant genes in FP17; Table S4: Antimicrobial susceptibility of *V. tubiashii* FP17. (Sensitivity: S = sensitive, M= moderately sensitive, R = resistant).

**Author Contributions:** Z.T., X.L. and C.D. conceived the study and designed the experiment; C.D., Z.T., X.L., D.L. and Q.L. operated animal issues; C.D. collected and analyzed the data; C.D., Z.T. and X.L. modified the figures and tables. C.D. wrote the manuscript. Y.S. revised the manuscript; M.S. was responsible for research activity planning and execution. We declare no competing interests. All authors have read and agreed to the published version of the manuscript.

**Funding:** This research was supported by the Hainan Province Key Research and Development Program, grant number ZDYF2022XDNY234 and the National Key Research and Development Program of China, grant number 2021YFC2600600.

**Institutional Review Board Statement:** The animal study was reviewed and approved by the Ethics Committee of Hainan Academy of Ocean and Fisheries Sciences (approval code HAOFS-2021-0012, approved on 9 August 2021).

**Informed Consent Statement:** The animal study was reviewed and approved by Ethics Committee of Hainan Academy of Ocean and Fisheries Sciences HAOFS-2021-0012 (approve on 9 August 2021).

**Data Availability Statement:** The data that support the findings of this study are available within the article, its Supplementary Materials, and from the corresponding author, Zhigang Tu, upon reasonable request.

**Acknowledgments:** We thank Zongsong Wu for reviewing the manuscript.

**Conflicts of Interest:** The authors declare to no conflict of interest.

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
