# Peer review of "First Report on Genome Analysis and Pathogenicity of Vibrio tubiashii FP17 from Farmed Ivory Shell (Babylonia areolata)"

_fishes, doi:10.3390/fishes7060396_

Round 1

Reviewer 1 Report

This is an interesting and important manuscript on vibrio infection from farmed ivory shells (Babylonia areolate. The study is well performed, the results are clearly presented and the manuscript is well written. I have just a few minor comments.

Line 25+26: a total of… were mapped; change to: a total of … was mapped

L 72: confirmed FP17 strain was the etiologial agent; change to: confirmed that FP17 was the etiological agent

L 113: was intramuscular injected; change to: was intramuscularly injected

L113: colony forming units (CFU)

L141: rRNA were; change to: rRNAs were

L206: chromosomesand; change to: chromosomes and

L243+244: although genome of FP17 was; change to: although the genome of FP17 was

L248: Additional neighbor-joining phylogenetic tree of FP17 strain were shown; change to: Additional neighbor-joining phylogenetic tree of FP17 strain was shown

L293: the common haemolysin toxin; change to: the common haemolysin toxins

L303: A total of 259 and 77 genes were mapped; change to: A total of 259 and 77 genes was mapped

L348: which suggests; change to: which suggest

L358: which indicates; change to: which indicate

L374: may linked; change to: may link

L380: Lpp play; change to: Lpp plays

L387: High concentration of the TP17 strain were found; change to: High concentration of the TP17 strain was found

L395: which leading to oyster death; change to: leading to oyster death

L414: but it dresisted to; change to. but it was resistant to

L416-417: A novel strain of…, where broke out large-scale acute death; change to: from a large-scale acute death

L417-419: Infectious experiment identified TP17 was; change to: Infectious experiment identified TP17 as

L420: The genome of FP17 strain is consists of; change to: The genome of FP17 consists of

Author Response

Dear reviewer,

     Thanks for your advices, the manuscript has been changed as you suggested. Please see the attachment. 

      Best ,

      Zhigang Tu

Reviewer 2 Report

improving in introduction and methods

Author Response

Dear reviewer,

       Thanks for your advices,we have changed the manuscript as you suggest. Please see the attachment.

Best,

Zhigang Tu

Reviewer 3 Report

The present manuscript a genomic analysis of a Vibrio strains assigned to the species V. tubiashii is described. However it is not clear that the studied strain was indeed V. tubisahii because the similarity of strain FP17 and ATCC 19019 based on 16S rRNA sequencing is only 94% which is too low to define species.

On the other hand, the manuscript resembles a "genome announcement" and therefore it has not been demonstrated  that any of the genes are actually associated with pathogenesis. This makes several sentences of the discussion speculative.

The ms need an editing of the English language 

Other specific questions:

- Figure 1 (A, B) must be eliminated

- Table 1 must include the physiological characteristics of different reference strains

- The results of pathogenicity of FP17 strains are confusing: in material and methods the bacterial concentration is expressed as CFU per snail and in results appears CFU/ml or CFU/ g body weight. Even in Fig. 4 the indications to which the concentrations refer are missing.

- line 344: Change pandemic for episode of mortality

-line 357: Flagellum and flagella (what is the difference?)

-line 388-389: is writted "The highest density of the colony was also observed with rearing tank water". Where it is done this in the manuscript?

- Line 164 and Change "paper diffusion" by "disk diffusion" 

- Several tables must be moved to Supplementary material (i.e., tables 3, 4)

Author Response

(The authors gave the same response as above.)

Round 2

Reviewer 3 Report

In the revised version of the manuscript the authors did not satisfy  adequatelly most of the previous comments or criticisms. 

-The presence of a large number of putative virulence genes does not necessarily imply that they are expressed in vivo and, therefore, that they play a role in the infectious process. To demonstrate this, it would be necessary to obtain the corresponding mutants. Then, the authors must eliminate several sentences in the abstract and in the discussion where it is indicated that several of the detected genes are responsible for certain infection processes. This had already been indicated in the first review.

- In the discussion (lines 357 & 358) is written" The moderate value of LD50 of FP17 may imply a key role of quorum sensing than virulence factor during the disease progression". This must be explained.

- The figure 3B showing the phylogenetic three is not cited within the text

- The differences among the physiological characteristics of the FP17 and the reference V. tubiashii strains should be cited in results and discussed  in the "Discussion section". What means -/+ in table 1? Maybe variable result or weak result?

- In line 110 (The infection experiment) is written ".. bacterial suspension with indicated concentration..." Where are the indicated concentrations??

- The results of pathogenicity assays  (lines 206 to 211) are very difficult to understand. All the paragraph must be rewritten. In addition, the authors  did not included within Fig.1, if the bacterial concentrations are CFU/g or CFU/ml as previously requested.

- The paragraph of discussion related to the differences among the authors in the LD50 values obtained by several authors are not relevant because some values are expressed as CFU/ml and others as CFU/g. In addition, the via of infection can be different (bath versus injection)

- The authors did not eliminate the term "pandemic" on line 336. 

- Fure 1 A and b must be eliminated (not transferred to supplementary material)

Author Response

Dear reviewer,

     Thanks for your advices, the manuscript has been changed as you suggested. Please see the attachment. 

      Sincerely ,

      Zhigang Tu

Round 3

Reviewer 3 Report

Many of the authors' responses are difficult to follow since the lines indicated where the changes were made do not correspond to the latest version of the manuscript.

Although all the manuscript must be reviewed by a native English speaker, some points where authors should pay special attention are indicated here:

- Line 20: change "of 14d" by "at 14d"

- line 28: change "was" by "were"

- Paragraph (lines 50-52) must be rewritten or eliminated. It is really confusing because it seems that the isolated strain has already been characterized before.

- Lines 73-74, must be rewritten as "In the current study a novel strain of V. tubiashii (named FP7) was isolated from moribund B. areolata farmed at Hainan during the past two yeras"

- Line  90  : the salinity was 30 .... ?(I suposs ppt ?)

- Lines 96 and 98: rewritte as " 2216 E agar (...) and Thiosulfate... (TCBS) (....) plates which were ....."

- line 110  : insert unit for salinity

-Line 111: "respectively to each group"

- Line 114  : Change "bacteria by "bacterium"

- Line 177: It is written "Single colony of different isolates". Why if only one strain was isolated?

- Line 180: change "papers" by "disks"

- Line 189: The authors must indicate the temperature range instead the maximum temperature of growth

- Line 206: Why "the dominant strains" if only a strain (FP7) was inoculated.

- Line 209  : change "of 14D" by  "at 14d"

- Lines 262-263: the comment is speculative and must be eliminated

- Line 336: the term "toxicity" is not employed for live cells. It must be changed by "virulence"

-Line 339: change "similar" by "quite similar"

-Line 348: change "was" by "can be assigned to the species V. tubiashii"

- Paragraph from line 352 to 356: It must be eliminated because the reference of Santos et al is related to bacterial fish diseases. Consequently, this reference must be also deleted from the reference list.

- line 374: Why "flagella" if the strain have a single polar flagellum

- Lines 384-385: The last sentence is speculative and must be deleted.

Author Response

Dear reviewer,

     Thanks for your advices, the manuscript has been changed as you suggested. Please see the attachment. 

      Best,

      Zhigang Tu
